# A Modified Ridge Splitting Technique Using Autogenous Bone Blocks—A Case Series

**DOI:** 10.3390/ma13184036

**Published:** 2020-09-11

**Authors:** Dorottya Pénzes, Fanni Simon, Eitan Mijiritsky, Orsolya Németh, Márton Kivovics

**Affiliations:** 1Department of Community Dentistry, Semmelweis University, 1088 Budapest, Hungary; simon.fanni@dent.semmelweis-univ.hu (F.S.); nemeth.orsolya@dent.semmelweis-univ.hu (O.N.); kivovics.marton@dent.semmelweis-univ.hu (M.K.); 2Head and Neck Maxillofacial Surgery, Department of Otolaryngology, Tel-Aviv Sourasky Medical Center, Sackler Faculty of Medicine, Tel-Aviv University, Tel Aviv 62431, Israel; mijiritsky@bezeqint.net

**Keywords:** mandibular ridge augmentation, piezo surgery, bone transplantation, alveolar bone loss, dental implantation

## Abstract

Background: Alveolar atrophy following tooth loss is a common limitation of rehabilitation with dental implant born prostheses. Ridge splitting is a well-documented surgical method to restore the width of the alveolar ridge prior to implant placement. The aim of this case series is to present a novel approach to ridge expansion using only autogenous bone blocks. Methods: Patients with Kennedy Class I. and II. mandibles with insufficient bone width were included in this study. Ridge splitting was carried out with the use of a piezoelectric surgery device by preparing osteotomies and after mobilization of the buccal cortical by placing an autologous bone block harvested from the retromolar region as a spacer between the buccal and lingual cortical plates. Block-grafts were stabilized by osteosynthesis screws. Implant placement was carried out after a 3-month healing period. A total of 13 implants were placed in seven augmented sites of six patients. Results: Upon re-entry, all sites healed uneventfully. Mean ridge width gain was 2.86 mm, range: 2.0–5.0 mm. Conclusions: Clinical results of our study show that the modified ridge splitting technique is a safe and predictable method to restore width of the alveolar ridge prior to implant placement.

## 1. Introduction

Alveolar atrophy following tooth loss is a common limitation of rehabilitation with dental implant born prostheses. Following tooth extraction, the jaw undergoes alveolar atrophy. The amount of bone reduction is more significant in the mandible compared to the maxilla [1]. There are notable differences observed in the rate and degree of atrophy between the anterior and posterior segments of the mandible, with the posterior segments showing a higher degree of bone loss [2]. Furthermore, the direction of the absorption also shows significant differences. Initially after tooth extraction, the mandibular bone loss is mainly horizontal, while vertical resorption is more significant at a later stage [3]. Inadequate bone width for dental implant placement in the premolar and molar regions of the mandible is common in a late implant placement protocol. For the long-term stability of the hard tissues, at least 1.5 mm of bone surrounding the implant on both oral and vestibular aspect is required [4].

There have been numerous techniques described for augmenting the mandible with deficient bone width. These techniques include onlay grafting using bone blocks, guided bone regeneration (GBR), shell technique, and the swinging interpositional grafts technique [5]. Each technique has its own disadvantages, with the most common complications being tissue dehiscence, membrane displacement or collapse, bone resorption, donor site morbidity in case of autografts used, long healing time, and inadequate quality of the augmented bone. Furthermore, all grafting methods significantly increase the morbidity of the patients, treatment cost, and time [6].

Numerous biomaterials, such as autologous bone, allografts, xenografts, alloplastic materials (i.e., nanomaterials), biologic mediators, and stem cells, have been successfully applied in various techniques to restore the horizontal and vertical dimensions of the alveolar ridges prior to or at the time of dental implant placement [7,8].

Tatum described the alveolar ridge splitting as a new way of alveolar ridge expansion [9]. The technique was later modified by Simion et al. [10] and Scipioni et al. [11].

Originally, the splitting is performed with chisel and hammer, later with rotating or oscillating saws [12,13].

The use of ultrasound for medical surgery facilitated a new method for ridge expansion. Piezoelectricity was discovered in 1880 by Jacques and Pierre Curie [14], and its use for surgery was explored in the 1940s [15]. It works with selective hard tissue cutting in a safe and precise way [16], while minimizing the risk of nerve or membrane injury because the soft tissues can oscillate at the same amplitude as the tip of the instruments [17]. It provides cleaner operation areas, and better visibility in the surgical field than rotary instruments [18]. Furthermore, the ultrasonic device enables the surgeon to carry out curved osteotomies [18]. However, poor efficacy and slow cutting rates of the piezosurgical instruments may be disadvantages of such a surgical method [19].

The aim of this case series is to present a novel approach to ridge expansion using only autogenous bone blocks.

## 2. Materials and Methods 

### 2.1. Patient Population

Patients who were periodontally healthy more than 18 years of age, with Kennedy Class I. and II. mandibles with insufficient bone width for implant placement, were included in our study.

Exclusion criteria were as follows:

History of uncontrolled medical disorders;History of systemic diseases or medication that alter bone metabolism;Poor oral hygiene;Smoking.

Criteria for patient selection were as follows:Mandibular ridge width at least 3 mm;Ridge height at least 11 mm;Spongiosa between the two cortical plates at least 1 mm [20].

The procedures involved in the study were thoroughly explained to the patients, and each patient gave informed consent in writing.

The study was approved by the Regional and Institutional Committee of Science and Research Ethics (52158-2/2015/EKU [0425/15]) and The Office of the Chief Medical Officer of The National Public Health and Medical Officer Service (IF-14561-10/2015). All investigations reported have been carried out in accordance with the Helsinki Declaration.

Seven ridge splitting procedures were performed in the mandible of six patients (one male). After the 3-month healing period, 13 implants were placed.

### 2.2. Surgical Procedure

Patients were required to rinse with a 0.2% chlorhexidine solution for 1 min before surgery. Under local anesthesia, a full-thickness flap was raised from a crestal incision with a mesial releasing incision to access the alveolar ridge and the retromolar area. A midcrestal osteotomy was preformed, leaving a safety zone of at least 2 mm from the adjacent tooth [13]. Two vertical releasing osteotomies were performed at the mesial and distal ends of the mid-crestal osteotomy. Apically, the vertical osteotomies were connected horizontally with a superficial corticotomy. Osteotomies and corticotomies were carried out using a piezoelectronic device (NSK Variosurg3 Ultrasonic Bone Surgery System, NSK Europe GmbH, Eschborn, Germany.) Chisels were inserted in the midcrestal osteotomy to create a green-stick fracture to allow extensive mobilization of the buccal cortical.

After mobilization of the buccal cortical, an autologous bone block with dimensions corresponding to that of the bone defect created in the recipient site was harvested from the retromolar area and was placed as a spacer between the buccal and lingual cortical plates. The block was stabilized using osteosynthesis screws (Meisinger Screw System, Hager and Meisinger GmbH, Neuss, Germany). The surgical steps are demonstrated in Figure 1. Both lingual and buccal flaps were mobilized to allow tension-free primary closure. The flap was closed with horizontal mattress sutures, and then single interrupted sutures closed the edges of the flaps. Suture removal took place after 14 days. All patients received amoxicillin and clavulanate (Aktil Duo 875 mg/125 mg, Sandoz Hungária Kft., Budapest, Hungary) 1 g twice per day, starting at the day of the surgery and continuing for 7 days. In case of amoxicillin allergy, clindamycin (Dalacin 300 mg, Pfizer Inc., New York, NY, USA) was prescribed four times a day for 7 days. Furthermore, a non-steroid anti-inflammatory drug, diclofenac (Cataflam 50 mg, Novartis Hungária Kft., Budapest, Hungary), 3 times a day for 3 days, and 0.2% chlorhexidine mouth rinse (Corsodyl, GlaxoSmithKline Consumer Healthcare GmbH & Co. KG, München, Germany), twice a day for 2 weeks, were prescribed to the patients. During the healing period, patients did not wear temporary prostheses.

Surgical re-entry took place after 3 months of healing. Implant bed preparation was carried out with rotatory instruments powered by a surgical micromotor (MasterSurg Surgical Systems, KaVo Dental Systems Japan, Co., Ltd., Tokyo, Japan). A trephine drill with an external diameter of 3.0 mm and an internal diameter of 2.0 mm (330 205 486 001 020 Hager and Meisinger GmbH, Neuss, Germany) with external cooling at a drill rotation speed of 800 rpm to the depth of 8 mm was used to remove bone core biopsy samples for histologic analysis. Implant beds were finalized according to the instructions of the implant manufacturer at a rotation speed of 800 rpm. Implants (Nobel Replace Conical Connection, Nobel Biocare AG, Kloten, Switzerland) were placed submerged in the augmented bone. Implant uncovery procedure took place 3 months after implant placement.

Patients were treated in outpatient care settings for minor surgeries without specific thromboprophylaxis prescribed. Patients were recalled the day following surgery to control postoperative bleeding.

All surgical interventions were carried out by the same operator (DP).

### 2.3. Clinical Measurement

Clinical measurements were carried out using Williams probe (Karl Hammacher GmbH, Solingen, Germany) prior to ridge splitting surgery after the full thickness flap elevation, to evaluate the width of the alveolar ridge (preoperative measurement) and after a 3-month healing period, before implant placement (postoperative measurement). The width of the alveolar ridges was measured at 3, 10, and 15 mm from the reference point, which was the distal marginal bone of the last tooth of the quadrant. To ensure that preoperative and postoperative measurement sites corresponded, we placed the tip of the Williams probe on the reference point described and the markings of the probe to the highest ridge of the lingual cortical.

## 3. Results

A total of six patients participated in the present study; median age was 56.5 years (range: 42–77 years). Seven staged ridge split procedures for lateral augmentation in the posterior mandible were performed. All sites showed improvement in the ridge width. There were notable differences in the ridge width gain at 3, 10, and 15 mm. Excellent ridge width gain was observed at 10 mm, mean ridge width gain was 2.86 mm range: 2.0–5.0 mm. At 3 mms and 15 mms, moderate width gain was achieved. Pre- and postoperative measurements are presented in Table 1.

All sites healed uneventfully, and upon re-entry excellent bone regeneration was observed with perfect ossification of the osteotomies, except for one site, which needed secondary augmentation. Clinically at the second surgery, the augmented area showed dense cortical bone at the coronal part of the ridge.

All augmented areas were sufficiently wide to accommodate implants according to the prosthetic plan. None of the implants failed at the 6-month control. The fixed dental prostheses were successful and functional in all cases.

The intra- and postoperative pain and swelling were comparable to other dentoalveolar surgeries. There were no cases of infection and no complications at the donor site. In one of the cases, the buccal cortical plate fractured during the ridge splitting procedure. The fractured plate was stabilized by osteosynthesis screws to the lingual cortex, and the healing was uneventful. In one of the cases upon reentry, connective tissue was observed between the buccal and lingual corticals. However, the block graft has carried out its space maintaining purpose. After careful debridement of the soft tissue, the implant bed was prepared. The implant placed in this site achieved primary stability and was anchored in the apical part of the ridge. The intrabony crater-like defect was regenerated with the guided bone regeneration technique (GBR) using bovine bone mineral matrix graft (creos xenogain, Nobel Biocare AG, Kloten, Switzerland) and nonresorbeable membrane (Permamem, Botiss biomaterials GmbH, Zossen, Germany.)

None of the patients in our study had a history of congenital bleeding disorders. None of the patients received anticoagulants; one of the patients was on antiplatelet medication. At the control appointments, no postoperative hemorrhage was observed.

## 4. Discussion

For the long-term success of dental implants, augmentation of the narrow alveolar ridge is necessary prior to implant placement. There are various approaches for lateral augmentation, i.e., onlay grafting using bone blocks [21], membrane protected blocks [22], GBR [23,24,25], and interpositional grafts [26].

Simion et al. described the defect created during the ridge splitting procedure as a “self-space making” defect [10], a four-wall intrabony box. This box can contain the graft materials better, than a one-wall defect observed at onlay grafting, or the GBR technique. Cortellini et al. described that the shape of the created defect is favorable for the formation of new vital bone [27].

The success of ridge expansion is not only dependent on the shape of the defect, but also on the grafting material used. Probably the most controversy surrounding the ridge splitting technique is the type of grafting material applied. In the literature autografts, xenografts and alloplastic materials were applied successfully in ridge splitting [6,12,13,26,28]. Furthermore, ridge expansion could be carried out without the use of graft materials. According to Gonzalez-Gracia et al., using a mixture of autogenous bone graft and allogenic bovine particulated bone graft showed predictable results in terms of bone regeneration [29]. According to histological results after the 4-month healing period the bone core biopsy specimens showed de novo formation of mature bone, with residual particles of the bone graft material [29].

Furthermore, Ramal et al. compared conventional ridge split with xenograft particulate covered by a collagen membrane with a modified bone expansion, in which the graft material was autogenous bone particles harvested from the ipsilateral surgical site. After six months of healing, there was no statistical difference in the crestal bone width between the two groups [30].

Other authors chose to use no graft material in the bone defect created by expansion. In these studies, dental implants were inserted during the ridge splitting surgery, to provide the function of a spacer to avoid the collapse of the bone defect [31,32,33]. The expanded site may be left empty, because it can be considered as a self-containing defect with four cortical walls, and complete bony regeneration is possible [34]. Scipioni et al. examined the micromorphology of the augmented areas following a graftless ridge splitting procedure using histological methods. After the 16-month healing period a mature, regenerated bone was observed [35].

Dottore et al. conducted vertical augmentation in the posterior mandible with interpositional autogenous bone graft. With this technique, the cortical bone was displaced coronally, where stable bone formation was observed, which contributed to the long-term stability of the inserted implants [5].

In their study, Mahmoud et al. treated 562 patients with a flapless piezotome crest split procedure using a synthetic self-hardening biphasic bone graft material. The baseline and final width of the alveolar ridge were 1.9 ± 0.4 mm and 6.5 ± 0.7 mm, respectively [36]. Flapless surgical methods allow for better blood supply and decreased marginal bone resorption, unlike full thickness flap preparation; however, our modified surgical approach required the elevation of a mucoperiosteal flap to allow the immobilization of the block grafts with the use of osteosynthesis screws [37,38].

Nickeninig et al. presented a novel approach for ridge splitting assisted by navigated surgery following virtual planning. Successive surgical templates were used to transport the virtually determined splitting vector, to guide the piezosurgical osteotomy, expansion screws, and spreading chisels, and to place the implants. Particulated autologous bone was used as filler in the bone defect created during the intervention [39].

In their study, Albanese et al. carried out ridge splitting using a piezosurgical device with simultaneous implant placement. Fresh frozen tissue bank allograft was used as filler in the bone defect covered by a double layer of membranes. Compared to autologous bone, allografts eliminate donor site morbidity; allografts are available in sufficient quantity, however, and higher costs and lack of vital osteoblasts are considered as their disadvantages [40].

In the present study, an autologous bone block harvested from the retromolar area was used to provide the function of a spacer. Autologous bone is considered the gold standard of bone graft materials due to its osteogenic, osteoinductive, and osteoconductive properties [41]. Donor site morbidity is considered a disadvantage of the use of autologous bone. However, in the surgical method described in this case study, donor and recipient regions are in close proximity and accessed from the same flap, which may decrease the postoperative discomfort of the patient.

The healing period in ridge splitting described in the literature varies between 4 and 6 months [26,28], with most techniques involving implant placement simultaneously [6,20,29,42]. The modified approach described in the present study consists of grafting with autologous bone blocks and second stage implant placement after 3 months. Depending on the implant system and loading protocol used, this means an overall healing period of 3–6 months. Longer healing time may be a weakness of this surgical method.

According to the review of Williams et al., venous thromboembolism is among the most common complications in inpatient healthcare settings [43]. Our modified technique was carried out in ambulatory settings without thromboprophylaxis prescribed, because according to the Official Journal of International Union of Angiology [44] and American College of Chest Physicians Evidence-Based Clinical Practice Guideline [45], the use of specific thromboprophylaxis is not recommended for low-risk general surgery patients who are undergoing minor procedures and have no additional thromboembolic risk factors. However, the use of piezosurgical instruments instead of rotational instruments increases the time of the surgery, where this prolonged time can manifest in a higher risk of extensive bleeding, especially in patients with congenital or acquired bleeding disorders [46,47]. Therefore, patients were closely monitored for postoperative hemorrhage.

Surgical instrumentation of ridge splitting procedures have undergone development since the technique was first introduced. The earliest ridge split procedures were carried out with chisels and hammers [10,13]. In the 2000s, rotating [13] and oscillating instruments appeared [48] for ridge splitting.

The use of the chisel and mallet may cause significant discomfort to the patient. Rotating instruments lower the patient’s stress level, but may lead to thinner, weakened cortical bone plates because of the relatively wide diameter of burs used.

Piezoelectric instruments provide a good alternative to the above-described methods because of the thin tips that enable the sparing management of bone. Further advantages of such a device include less patient discomfort, minimizing the risk of soft tissue or nerve injury, and facilitated surgical accuracy by allowing preparation of curved osteotomies. However, slow cutting rates and increased surgical time may be a disadvantage of piezoelectrical instruments [16,17].

## 5. Conclusions

Within the limitations of the present study, we have concluded that the modified ridge splitting technique using autologous bone block grafts is an effective method to restore the width of the alveolar ridge prior to implant placement.

## Figures and Tables

**Figure 1 materials-13-04036-f001:**
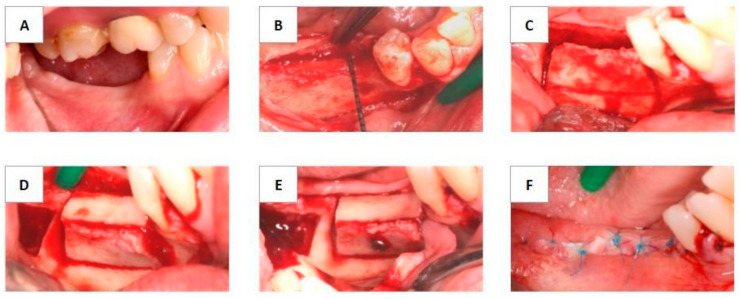
Clinical illustration of the surgical procedure of the modified ridge splitting. (**A**) Preoperative view of the atrophied alveolar ridge. (**B**) Full thickness flap preparation. (**C**) Buccal view of the osteotomies and the corticotomy. (**D**) The autologous bone block graft placed in the recipient site. (**E**) The graft was stabilized with osteosynthesis screws. (**F**) Tension free primary closure.

**Table 1 materials-13-04036-t001:** Pre- and postoperative measurements of the alveolar ridges, and the characteristics of the inserted implants (A—Need for secondary augmentation, B—Buccal cortical plate fractured during the ridge splitting procedure) (MR—right side of the mandible, ML—left side of the mandible).

Patients	AGE	SEX (M = Male, F = Female)	Surgical Area	Preoperative Measurement			Postoperative Measurement			Ridge Width Gain			Position of the Implants	Diameter and Length of the Implants (mm)
				3 mm	10 mm	15 mm	3 mm	10 mm	15 mm	3 mm	10 mm	15 mm		
Patient 1	56	M	ML	5.0	4.0	5.0	5.0	8.0	6.0	0.0	4.0	1.0	36	3.5 × 10
													37	4.3 × 10
			MR	5.0	3.0	4.0	6.0	5.0	5.0	1.0	2.0	1.0	44	3.5 × 10
													46	3.5 × 8
Patient 2 ^A^	77	F	MR	7.0	4.0	5.0	9.0	9.0	9.0	2.0	5.0	4.0	45	3.5 × 10
													46	3.5 × 10
Patient 3 ^B^	44	F	MR	5.0	5.0	7.0	6.0	8.0	9.0	1.0	3.0	2.0	45	3.5 × 10
													46	4.3 × 10
Patient 4	57	F	ML	4.0	7.0	6.0	8.0	9.0	7.0	4.0	2.0	1.0	36	4.3 × 8
Patient 5	70	F	MR	4.0	4.0	4.0	5.0	6.0	7.0	1.0	2.0	3.0	45	3.5 × 8
													46	3.5 × 8
Patient 6	42	F	MR	4.0	6.0	5.0	7.0	8.0	8.0	3.0	2.0	3.0	44	4.3 × 11.5
													46	4.3 × 11.5
Mean	-	-	-	4.86	4.71	5.14	6.57	7.57	7.29	1.71	2.86	2.14	-	-

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
