# Peer review of "A Modified Ridge Splitting Technique Using Autogenous Bone Blocks—A Case Series"

_materials, 2020, doi:10.3390/ma13184036_

Round 1
Reviewer 1 Report
INTRODUCTION:
At the end of the introduction, the reason for writing the manuscript is usually briefly presented.
DISCUSSION:
I think that the present manuscript will benefit from a wider literature discussion, including some surgical techniques which were not considered in the Discussion section.
The paper of Mahmoud et al. presented a sample of 567 patients treated with autologous bone block grafting (ABBG) and 562 treated with flapless piezotome crest split (FPCS). The baseline crest width was around 2 ± 0.5 mm and the final crest width achieved with ABBG and FPCS was 5.8 ± 0.8 mm and 6.5 ± 0.7 mm, respectively, with a mean gain of 4 – 4.5 mm. The sample of 6 patients presented in this manuscript showed a mean preoperative ridge width of 4.8 mm and a mean postoperative ridge width gain of 2.86 mm. When presenting a modified surgical technique, results should be compared with existing techniques.
The paper of Nickenig et al. presented an interesting ridge splitting technique using surgical guides for ridge osteotomy and expansion, without bone augmentation and with immediate implant placement. Major advantages of this technique are the reduction of clinical intervention and healing time, together with the absence of donor site morbidity. The patient presented in this case report had a horizontal alveolar ridge width of 4 mm, similar to the sample presented in this manuscript: which is another alternative to be compared with.
The paper of Albanese et al. presented a sample of 10 patients undergoing horizontal bone augmentation through the use of tissue bank fresh frozen chips and double-layer collagen membranes. A comparison could be made between cost, harvested bone volume, final alveolar bone width and healing process of this technique and the technique presented by the authors of the present manuscript.
REFERENCES:
References format should be modified according to journal guidelines.
Reviewer 2 Report
The manuscript presented a ridge splitting technique for implant placement .However , the creation is not enough for this technique. As a case series , 6 cases are not strong for supporting , and the 3 month follow up is too short to observe the clinical outcome for ridge splitting . The statistic analysis is important for a case series , we didn't find the analysis in this manuscript .
There are some points need to improve :
1. line 97 -98 , there no company information for the medication .
2. line 109 , the measurement is not clear . The reproduction must be considered in the measurement .
3. line 137 , vertical augmentation was described . It is very strange .
Reviewer 3 Report
Dear colleagues,
unfortunately no quantitative data have been presented (like histomorphometrical analysis) to substantiate your hypothesis. Furthermore, no control group was included.
Thus, the manuscript cannot be published.
Best wishes
Reviewer 4 Report
The authors presented an interesting topic related to approach to ridge expansion using only autogenous bone blocks. The topic is interesting, however the manuscript requires several corrections and additions:
In the introduction the authors should also mention the alternative materials used in the regeneration of the alveolar bone, e.g Materials 2015, 8, 2953-2993; doi:10.3390/ma8062953; Nanomaterials 2020, 10, 1216; doi:10.3390/nano10061216.
Whether all the procedures were performed by the same operator?
What was the diameter of the used implants and drills?
Did the authors apply an external/internal cooling during preparation of implant bed?
What type of dental implant micromotor was used?
What was the drill rotation speed?
References need updating.
Reviewer 5 Report
The authors of the manuscript focused on ridge splitting technique using autogenous bone blocks in a total of 13 implants, who were placed in 7 augmented sites of 6 patients
In part the materials and methods , it is necessary that the authors describe the monitoring of hemostasis postoperatively. It is not enough to write that these were patients older than 18 years, but a median age is also needed. The whole manuscript is very transparent and fills with new knowledge that is important in oral surgery.
Tables and figures in the text are very clearly written.
In the results, the authors should describe more the presence or absence of bleeding.
The discussion needs to be extended to the following facts.
Describe and compare postoperative complications, bleeding or thrombosis with other available studies. I would also like to explain if the patients were set up for some anticoagulant treatment in the postoperative period. Venous thromboembolism was the second most common medical complication, the second most common cause of increased length of hospital stay, the third most common cause of mortality and a significant increase in financial cost. Williams B et al. J Oral Maxillofac Surg. 2011 Mar;69(3):840-4.
The authors describe that increased surgical time may be a disadvantage of piezoelectrical instruments. It is very difficult to manage patients with congenital bleeding disorders (fibrinogen disorders, von Willebrand disease) that require careful attention. It is necessary to emphasize the perioperative management of the patient with bleeding disorders. It is in these patients that dental procedures are often required due to frequent mucosal bleeding. Treatment should be adjusted in these patients to prevent bleeding or thrombosis. It is also appropriate to quote this publications : Simurda T et al. Perioperative Coagulation Management in a Patient with Congenital Afibrinogenemia during Revision Total Hip Arthroplasty. Semin Thromb Hemost. 2016 Sep;42(6):689-92. doi: 10.1055/s-0036-1585079. and Simurda et al. Successful Use of a Highly Purified Plasma von Willebrand Factor Concentrate Containing Little FVIII for the Long-Term Prophylaxis of Severe (Type 3) von Willebrand's Disease. Semin Thromb Hemost. 2017 Sep;43(6):639-641.doi: 10.1055/s-0037-1603362.
I have to say that with these 36 references there are only 6 references newer than 5 years old. Therefore, it is necessary to add newer references.
The conclusion is short but concise
Round 2
Reviewer 2 Report
The authors improved this manuscript , but the improvement is not enough.
Most of points in my last review were not improved. As a case report , creation , and long time follow up are very important to be considered. As a case series ,
statistic analysis is important. I can't see these points in this manuscript. Therefore , I suggest to reject for publication . However , I also encourage the authors to resubmit this manuscript with longer follow up data (at least 1 year ) and more cases.
Reviewer 3 Report
Dear colleagues,
many thanks for the revisions. In my eyes your manuscript is now suitable for publication.
Best wishes
Reviewer 5 Report
The presented manuscript has been corrected in response to the suggestions. The authors have followed the recommendations of the reviewers. After the revision, the provided data and interpretation of the results became more clear. I would like to thank the authors for resubmitting the manuscript and explaining the obscure points from the previous version.
Now, the revised manuscript can be accepted for publication.